# Contemporaneous Perioperative Inflammatory and Angiogenic Cytokine Profiles of Surgical Breast, Colorectal, and Prostate Cancer Patients: Clinical Implications

**DOI:** 10.3390/cells12232767

**Published:** 2023-12-04

**Authors:** Leili Baghaie, Fiona Haxho, Fleur Leroy, Beth Lewis, Alexander Wawer, Shamano Minhas, William W. Harless, Myron R. Szewczuk

**Affiliations:** 1Department of Biomedical & Molecular Sciences, Queen’s University, Kingston, ON K7L 3N6, Canada; 16lbn1@queensu.ca (L.B.); fiona.haxho@mail.utoronto.ca (F.H.); fleur.leroy@etu.unistra.fr (F.L.); 2Dermatology Residency Program, the Cumming School of Medicine, University of Calgary, Calgary, AB T2T 5C7, Canada; 3Faculté de Médecine, Maïeutique et Sciences de la Santé, Université de Strasbourg, F-67000 Strasbourg, France; 4ENCYT Technologies Inc., Membertou, NS B1S 0H1, Canada; blewis@encyt.net (B.L.); alexwawer@gmail.com (A.W.); shamano12@hotmail.com (S.M.)

**Keywords:** surgery-induced wound healing, growth factors, chemokines, cytokines, surgical breast, prostate, colorectal cancer patients, pro-inflammatory and angiogenic plasma profiles of cancer patients

## Abstract

Surgery-induced tumor growth acceleration and synchronous metastatic growth promotion have been observed for decades. Surgery-induced wound healing, orchestrated through growth factors, chemokines, and cytokines, can negatively impact patients harboring residual or metastatic disease. We provide detailed clinical evidence of this process in surgical breast, prostate, and colorectal cancer patients. Plasma samples were analyzed from 68 cancer patients who had not received treatment before surgery or adjuvant therapy until at least four weeks post-surgery. The levels of plasma cytokines, chemokines, and growth factors were simultaneously quantified and profiled using multiplexed immunoassays for eight time points sampled per patient. The immunologic processes are induced immediately after surgery in patients, characterized by a drastic short-term shift in the expression levels of pro-inflammatory and angiogenic molecules and cytokines. A rapid and significant spike in circulating plasma levels of hepatocyte growth factor (HGF), interleukin-6 (IL-6), placental growth factor (PLGF), and matrix metalloproteinase-9 (MMP-9) after surgery was noted. The rise in these molecules was concomitant with a significant drop in transforming growth factor-β1 (TGF-β1), platelet-derived growth factor (PDGF-AB/BB), insulin-like growth factor-1 (IGF-1), and monocyte chemoattractant protein-2 (MCP-2). If not earlier, each plasma analyte was normalized to baseline levels within 1–2 weeks after surgery, suggesting that surgical intervention alone was responsible for these effects. The effects of surgical tumor removal on disrupting the pro-inflammatory and angiogenic plasma profiles of cancer patients provide evidence for potentiating malignant progression. Our findings indicate a narrow therapeutic window of opportunity after surgery to prevent disease recurrence.

## 1. Introduction

Surgical tumor removal remains the primary curative treatment option for cancer patients. Unfortunately, decades of research and clinical reports have established that surgery can also promote the growth of residual tumor cells and accelerate metastatic spread [1,2]. Resection of a primary tumor can promote disease progression in several ways. It is known that surgical intervention can induce the shedding of tumor cells into the circulation, a phenomenon that has been clinically observed during the resection of lung [3,4], colorectal [5,6], breast [7,8,9], esophageal [10], prostate [11], gastric [12], and pancreatic cancer [13]. As a result, the subsequent local and systemic immunologic changes accompanying surgery may directly affect disseminated tumor cells in circulation, particularly the wound-healing response [14,15,16]. This process is initiated by a pro-inflammatory phase characterized by a coagulation cascade and platelet activation. It has been shown to promote the survival and metastasis of circulating tumor cells (CTC) via several mechanisms. Firstly, platelet activation triggers the release of platelet-derived growth factor (PDGF-AB/BB), which can directly act on tumor cells, endothelial cells, fibroblasts, and other stromal cells. PDGF-AB/BB binding to CTCs can stimulate cell growth and proliferation and increase the expression of cell surface adhesion molecules that facilitate CTC attachment to microvasculature [17,18,19]. This process is accelerated by the increased activated platelet levels, which can bind CTCs and facilitate their ability to adhere to endothelial cells [20] stably. Furthermore, upregulated PDGF-AB/BB levels stimulate tumor neovascularization, a process that not only supports tumor growth but also provides a means for the entry of tumor cells into the circulation and distant metastasis [21].

CTCs can also survive in circulation by binding to various coagulation factors markedly produced during the pro-inflammatory phase of wound healing. These factors include fibrinogen, fibrin, tissue factor, and thrombin [22,23]. The binding of activated platelets and coagulation factors to cancer cells results in the formation of a tumor cell embolus that can evade immune detection [24] and undergo growth arrest in capillary beds before tissue extravasation [25]. Notably, tumor cells can also secrete many of these coagulant factors themselves to activate the coagulation cascade [26], enhance their survival in circulation [27], and stimulate tumor vascularization [28]. The binding of activated platelets to the endothelium has also been implicated in developing the pre-metastatic niche [29,30]. This process is instigated by the secretion of stromal cell-derived factor-1 (SDF-1 or CXCL12), a chemokine that recruits bone marrow-derived progenitor cells to areas of inflammation. CTCs commonly expressing CXCR4, the cognate receptor for SDF-1, may also be mobilized and recruited into the pre-metastatic niche [31]. The role of coagulation and platelet activation in tumor progression implicates the importance of targeting the wound-healing process, particularly during the narrow perioperative period following tumor resection. Indeed, anti-coagulants such as heparin or antiplatelet inhibitory antibodies have shown promising effects in the adjuvant setting by their ability to suppress metastasis and tumor-associated angiogenesis [32]. Unsurprisingly, similar well-tolerated drugs, such as acetylsalicylic acid (Aspirin^©^), are increasingly prescribed after surgical interventions [33,34,35].

Following the inflammatory phase of wound healing, the proliferation phase is initiated and characterized by the activation and proliferation of fibroblasts, endothelial cells, and immune cells and the concomitant secretion of potent stimulatory growth factors. These include epidermal growth factor (EGF), fibroblast growth factor (FGF), vascular endothelial growth factor (VEGF), and hepatocyte growth factor (HGF), each of which has been implicated in the malignant progression of various cancer types [36]. During the proliferation phase, these growth factors can instigate various tumorigenic processes, including anchorage-independent survival of disseminated CTCs, proliferation of fibroblasts that synthesize collagen in the tumor stroma, and proliferation of endothelial cells that comprise tumor-associated vasculature [36]. The development of this vascularization, induced by the aforementioned angiogenic signaling factors and hypoxia, provides the nutrients required for tumor growth and metastatic dissemination [21].

Besides wound healing, removing the primary tumor can promote local recurrence and malignant progression in several other ways. Early studies by Fisher and colleagues have demonstrated that the primary tumor and distant micrometastases are communicating systems in which manipulation of one system has significant consequences on the other [37,38,39,40]. They showed that when two tumors were inoculated simultaneously in both hind legs of mice, removing one tumor-bearing leg resulted in the accelerated growth of the remaining tumor and subsequent pulmonary metastases. The same group also conducted sham surgeries to demonstrate that this phenomenon was not solely due to the wound-healing process [38]. They found that resection of a non-tumor-bearing front leg did not enhance the growth of tumors or promote metastasis—a tumor mass had to be removed. An important mechanism explaining these observations was proposed in 1994 by Folkman et al., who demonstrated that a primary tumor can secrete angiostatin, a molecule that actively suppresses angiogenesis in distant tissues, or in other words, micrometastases [41,42]. Angiostatin was later found to inhibit endothelial cell proliferation in vitro by binding to ATP synthase, integrins, and annexin II [43]. More recent studies report similar effects of primary tumor resection on angiostatin levels and subsequent growth, neovascularization, and invasion of tumor cells [44,45].

Interestingly, Fisher et al. also highlighted the importance of timing for postoperative adjuvant treatment. Using animal models, they found that after removing a primary mammary tumor, the proliferation rate of metastatic cancer cells increased dramatically and could be detected within 24 h of tumor removal [38,39]. Metastatic proliferation peaked after 48–72 h; however, cell viability was reportedly sustained until the study’s endpoint. Notably, the study indicated that cyclophosphamide (CP) therapy was maximally effective when administered prior to or on the same day as tumor removal and that the efficacy of CP significantly decreased if treatment was delayed 3 or 7 days postoperatively [38,39].

These reports, as well as accumulating evidence from retrospective clinical studies, implicate a critical window of opportunity after surgery during which adjuvant treatment may be initiated to ameliorate the malignance-promoting effects of surgery and to improve overall patient outcomes—the perioperative period. Here, we investigate the effects of surgical interventions on the inflammatory and immunological response of patients with breast, colorectal, and prostate cancer. Plasma sampling was taken immediately before and after surgery and after 24 h, 48 h, 72 h, 1 week, 2 weeks, and 4 weeks post-surgery. These time points comprise the perioperative period and the physiological phases associated with surgical wound healing [46,47]. Importantly, we examine the plasma profile during this critical period and note trends in the circulating levels of pro-inflammatory and angiogenic cytokines, chemokines, and growth factors. Using multiplexed immunoassays, our data detect and quantify 19 significant mediators of inflammation, wound healing, and vascularization. We hypothesized that the transient upregulation of these plasma factors, particularly the potent mitogenic ligand HGF and pro-inflammatory cytokine IL-6, may directly result from surgical intervention and, thus, the primary mechanism by which surgery influences malignant growth. We also think that the initial drop in TGF-β immediately after surgery and subsequent increase to baseline may also be significant in this process. We reveal for the first time a detailed and extensive characterization of the plasma inflammatory and immune profile of surgical cancer patients during the perioperative period. These findings suggest that there may be a very narrow window of time after surgery to optimize our ability to eradicate any surviving cancer cells and to improve our ability to cure cancer. Suppose our hypothesis is correct that the perioperative period is when cancer cells may have an enhanced ability to metastasize; in that case, many people are dying today of potentially curative cancer because of delays in the initiation of treatment after surgery to allow for wound healing.

## 2. Materials and Methods

### 2.1. Patient Study

Plasma samples from 68 cancer patients from Cape Breton Regional Hospital, Nova Scotia, Canada, were analyzed: 31 breast, 26 colorectal, and 11 prostate cancer patients. Blood Collection staff at the Cape Breton Regional Hospital collected blood samples just before the patients’ surgeries and during the patients’ post-surgery hospital stays. Encyt staff collected the blood samples after the patients had been discharged to go home. All samples were taken to the laboratory at Encyt Technologies Inc. for processing immediately after collection. All patients were controlled for treatment received and had not received neoadjuvant therapy (chemotherapy or radiation) before surgery for primary tumor removal. Eight time points were analyzed: pre-surgery (collected within 4 h before surgery), and 2 h, 24 h, 48 h, 72 h, 1 week, 2 weeks, and 4 weeks post-surgery.

The Nova Scotia Health Research Ethics Board approved this study. All patients enrolled in the study provided informed consent.

### 2.2. Multi-Plex Magnetic Bead Panel Kits

EMD Millipore’s (Burlington, MA, USA) MILLIPLEX MAP Human Cytokine/Chemokine (Cat: HCYTOMAG-60K), Human Angiogenesis/Growth Factor (Cat: HAGP1MAG-12K), TGFβ1 Single Plex (TGFBMAG-64K-01), Human Cytokine/Chemokine Panel II (HCYP2MAG-62K), Human MMP (HMMP2MAG-55K), and Human IGF-1, II (HIGFMAG-52K) Magnetic Bead Panel kits were used for the quantification of the following 19 human cytokines, chemokines, and growth factors: insulin-like growth factor-1 (IGF-1), transforming growth factor-β (TGF-β), interleukin (IL)-1α, IL-1β, IL-6, IL-8 (CXCL8), monocyte chemoattractant protein-1/chemokine ligand-2 (MCP-1/CCL2), MCP-2 (CCL8), tumor necrosis factor-α (TNF-α), vascular endothelial growth factor (VEGF), epidermal growth factor (EGF), stromal-derived factor-1αβ (SDF-1αβ), matrix metalloproteinase-9 (MMP-9), hepatocyte growth factor (HGF), placental growth factor (PLGF), platelet-derived growth factors (PDGF-AA, PDGF-AB/BB), leukemia inhibitory factor (LIF), and fibroblast growth factor-2 (FGF-2).

### 2.3. Preparation of Plasma Samples and Test Reagents

Whole blood using EDTA as an anti-coagulant was centrifuged at 2490 RCF for 10 min to isolate plasma. After initial centrifugation and aliquoting into polypropylene tubes, plasma was recentrifuged and then aliquoted into eight labeled polypropylene microfuge tubes and frozen at −80 °C. Samples from all collections were kept at −80 °C in a separate box for each patient until all samples were received from the patient. No samples were assayed immediately—samples with multiple (>2) freeze/thaw cycles were not assayed. Before testing, samples were thawed, vortexed, and centrifuged to remove particulates.

### 2.4. Immunoassay Procedures

Analysis was performed according to instructions in each of the EMD Millipore Magnetic Bead Panel kits used. The plates were assayed on Luminex 200™ with xPONENT 3.1 software. The Median Fluorescent Intensity (MFI) data were analyzed and recorded using a linear or 5-parameter logistic best-fitting curve for calculating analyte concentrations in each well. The calculated concentration was multiplied by the dilution factor for diluted plasma samples.

### 2.5. Statistical Analysis

Statistical analyses of data were performed using GraphPad Prism 10.0 software. Results were compared using one-way analysis of variance (ANOVA) at 95% confidence using Fisher’s LSD test.

## 3. Results

### 3.1. Plasma Expression of MMP-9 and Growth Factors Levels during the Perioperative Period

One of the earliest physiological changes detected in patients was the significant increase in matrix metalloproteinase 9, reaching peak levels within two hours after surgery (Figure 1). It is well known that MMP-9 plays essential roles in matrix remodeling, tissue damage repair, and neovascularization. Thus, it logically follows that MMP-9 levels would increase due to surgical wound healing [48,49,50]. Notably, the postoperative rise in MMP-9 was consistent among all cancer types (Figure 1A), although to varying degrees (Figure 1B–D) and returned to baseline 24 h after surgery. This suggests that the early induction of MMP-9 is short-lived yet may be a prerequisite for the host immunologic response that follows.

Plasma placental growth factor (PLGF) was also analyzed (Figure 2). PLGF is an angiogenic molecule essential for normal growth and vasculogenesis during embryonic development. Importantly, PLGF is also well documented for its ability to promote tumor progression [51,52,53]. Our findings indicate that plasma PLGF levels significantly increase in prostate cancer patients within 24–48 h after surgery (Figure 2D); however, similar patterns are not observed in breast (Figure 2B) or colorectal cancer patients (Figure 2C). Although these findings seem to implicate cancer-type-specific induction of PLGF during the perioperative period, additional studies may be required to elucidate these potential mechanisms.

Plasma growth factor levels were analyzed in surgical cancer patients, including hepatocyte growth factor (HGF), epidermal growth factor (EGF), and insulin-like growth factor-1 (IGF-1). We observed a rapid and significant rise in HGF levels after surgery, reaching maximal levels on postoperative day (POD) 1 for breast and colorectal cancer patients and on POD3 for prostate cancer patients (Figure 3B–D). Pooled data for all patients indicated a significant ~4-fold increase in HGF levels 24 h after surgery (Figure 3A). Similar to other plasma factors studied, HGF levels normalized after one week. Although there is variation between different patient groups, each cancer type demonstrated a similar postoperative trend for HGF. In particular, colorectal cancer patients exhibited the most significant increase in plasma HGF despite having comparable pre-surgery levels (Figure 3D).

In contrast, we also observed a modest decrease in the expression of EGF, consistent among different cancer types (Figure 4). IGF-1 levels were also markedly reduced postoperatively (Figure 5). For colorectal cancer patients, levels of IGF-1 appeared to decline steadily after surgery and reached minimum concentrations by POD3 before returning to baseline (Figure 5C). Our data suggest that breast cancer patients maintain significantly higher concentrations of IGF-1 during the perioperative period (Figure 5B). Beyond the time frame of the present study, the treatment outcome of these patients remains unknown. Thus, the clinical significance of these findings has not yet been evaluated to determine a possible link between perioperative immunologic changes and their potential clinical consequences.

### 3.2. The Pro-Inflammatory Plasma Cytokine Profile following Surgical Tumor Removal

Our findings indicate that surgical wound-healing is accompanied by a striking, yet short-lived, immunologic response mediated by IL-6 (Figure 6), and moderate changes in the expression of IL-1α (Figure 7), IL-1β (Figure 8), tumor necrosis factor-α (TNF-α; Figure 9), leukemia inhibitory factor (LIF; Figure 10), and transforming growth factor-β1 (TGF-β1; Figure 11) in plasma. Unsurprisingly, we detected significantly elevated levels of pro-inflammatory cytokine IL-6 in all patients shortly after surgery. IL-6 was found to be steadily increasing and reached peak plasma levels on POD1 for almost all patients analyzed. After this time, IL-6 levels began to drop back to pre-surgery levels after 72 h for most breast and colorectal patients and after one week for all other patients (Figure 6). Like HGF, the rise in IL-6 was profoundly more significant for colorectal cancer patients than breast or prostate cancer patients (Figure 6C,D). It has been suggested that the increased production of pro-inflammatory cytokines in the plasma, particularly IL-6, is directly associated with the invasiveness of the surgical procedure [54,55,56,57,58,59,60]. Thus, the drastic IL-6 surge observed for colorectal cancer patients may reflect the magnitude of the inflammatory response initiated during surgery [54,55].

Plasma profiles of IL-1α (Figure 7) and IL-1β (Figure 8) during the perioperative period exhibited high variation between patients and did not convey a discernable trend among specific cancer types. However, pooled data from all patient samples analyzed indicated a modest reduction in both IL-1α and IL-1β 24 h after surgery before returning to baseline values within two weeks. Analysis of TNF-α also showed high variation among patients (Figure 9), and additional studies may be necessary to profile the perioperative expression of TNF-α. Notably, plasma levels of LIF significantly decreased almost 2-fold after surgery (Figure 10).

TGF-β1 was also analyzed (Figure 11). All patients exhibited a sharp decrease in plasma TGF-β1 level postoperatively, reaching minimum values regarding POD1 for breast and colorectal cancer patients (Figure 11B,C) and POD2 for prostate cancer patients (Figure 11D) before returning to baseline within one week. Notably, colorectal cancer patients demonstrated significantly higher TGF-β1 levels at the study’s endpoint than in the preoperative period. Pooled patient data indicated an approximately 2-fold decrease in TGF-β1 between 2 and 24 h after surgery. Although similar patterns emerge for breast, colorectal, and prostate cancer patients, it is clear that TGF-β1 levels vary significantly between cancer types. As previously described for IL-6, this variation may be linked to the invasiveness of different surgical procedures and the subsequent immunologic response that occurs [55].

### 3.3. Surgery-Induced Wound Healing and Angiogenesis Are Mediated by Plasma Chemokines

SDF-1α/β plasma levels (also called CXCL12) were also analyzed (Figure 12). SDF-1α/β levels progressively decreased after surgery, demonstrating the lowest concentrations of POD1 before returning to baseline within one week for most patients. A comparable decrease was observed in the levels of monocyte chemoattractant protein-2 (MCP-2), also called chemokine cytokine ligand-8 (CCL8) (Figure 13). Both SDF-1 and MCP-2 are substantial chemotactic factors responsible for recruiting immune cells to areas of tissue injury or inflammation. It was suspected that SDF-1 and MCP-2 plasma levels would increase following surgery due to their involvement in wound healing. However, it has been reported that their downregulation in the plasma directly results from their localized release in wounded tissue, where they can recruit leukocytes and other immune cells and promote angiogenesis [61,62]. The transient plasma profile of MCP-1 (CCL2) was less clear. MCP-1 levels fluctuated significantly after surgery between each patient group (Figure 14). We also investigated the plasma levels of IL-8, alternatively called CXCL8 (Figure 15), an essential angiogenic chemokine involved in wound healing. In colorectal cancer patients, it has been previously reported that IL-8 levels rise after surgery, concomitant with the rapid IL-6 surge [63,64,65,66]. Although a subset of patients analyzed in the present study demonstrated an increase in IL-8 after surgery, cumulative plasma analyses indicated high variation, and no significant perioperative trend was observed. Due to the critical role of inflammatory and angiogenic chemokines in immune-mediated tumor progression, additional studies may be helpful to validate these findings.

### 3.4. The Perioperative Period Is Characterized by a Drop in Plasma Pro-Angiogenic Growth Factors

The plasma concentrations of several angiogenic signaling molecules and growth factors were also determined. Our findings indicate that the levels for many of these angiogenic factors, specifically FGF-2, PDGF-AB/BB, and, to a lesser extent, VEGF, decreased after surgery and reached minimum levels by POD1. Pooled patient analyses indicated compared to VEGF (Figure 16A) that plasma FGF-2 (Figure 17A), PDGF-AA (Figure 18A), and PDGF-AB/BB (Figure 19A) levels significantly decreased approximately 2-fold within only 24 h, and to a lesser extent for VEGF (Figure 16A). Notably, plasma levels of PDGF-AB/BB are significantly higher than PDGF-AA at every sampling period measured. This is consistent with other reports describing that in comparison to PDGF-AB/BB, PDGF-AA is significantly less efficient at binding plasma proteins in the blood and is thus more rapidly degraded in circulation [67].

The postoperative drop in FGF-2 and PDGF-AB/BB ligands within 24 h may be multi-factorial. Generally, a significant drop in angiogenic factors is a direct result of primary tumor removal along with its associated stromal tissue, fibroblasts, and immune and endothelial cells that collectively work to produce these factors during tumor growth, tissue repair, and wound healing. Perhaps what may be more clinically relevant is the consequent rapid rise observed in FGF-2, PDGF-AB/BB, and VEGF within one week after surgery. Notably, this process significantly surpasses baseline levels for both FGF-2 and PDGF-AA in colorectal cancer patients (Figure 17C and Figure 18C). It is also noteworthy that plasma levels of PDGF-AA and PDGF-AB/BB were significantly higher in breast cancer patients (Figure 18B and Figure 19B) than in prostate cancer patients (Figure 18D and Figure 19D). Since stromal PDGFα- and β-receptor expression is a common and often upregulated characteristic of solid breast cancer tumors [68,69], elevated PDGF-AB/BB in the plasma may have significant consequences for triggering local recurrence or distant metastasis. Indeed, PDGFβ receptor (PDGFβR) overexpression on mammary tumor cells correlates with a less favorable prognosis after surgery [69,70].

## 4. Discussion

Although surgical removal of the primary tumor provides the best chance of long-term disease-free survival, patients are at high risk for the development of metastases after successful resection of the primary tumor. It is becoming increasingly clear that surgery can contribute to developing local recurrence and distant metastases. Surgical tissue trauma and concomitant wound-healing processes induce a systemic response that can promote residual tumor cells’ survival, proliferation, and metastatic growth and stimulate angiogenesis. Here, we reveal the narrow perioperative timeframe characterized by a drastic shift in the plasma profile of breast, colorectal, and prostate cancer patients.

One of the earliest changes observed was the rapid increase in plasma MMP-9, reaching peak levels two hours after surgery. The significant upregulation of plasma MMP-9 during the initial inflammatory phase of wound healing is consistent with the known role of MMP-9 in matrix remodeling and its involvement in epithelial tissue repair. Other studies have shown that commonly administered anesthesia during tumor resection also contributes to the enhanced production of MMP-2 and -9, reflective of serum concentrations [71,72,73,74,75]. Kirman et al. have also shown that the release of MMPs following tumor resection can promote the invasion capacity and motility of CTCs via MMP-9-mediated degradation of collagen IV in the basement membrane or the ECM [76]. Furthermore, elevated MMP-9 in the tumor microenvironment stimulates neovascularization and tumor progression.

Angiogenic growth factors analyzed, including VEGF, FGF2, TGF-β1, PDGF-AA, and PDGF-AB/BB, each demonstrated a notable decrease in plasma levels during the perioperative period. When compared to pre-surgery values, the pooled patient analysis indicated an approximate 30% decrease in VEGF levels 48 h after surgery, a 42% decrease in FGF2 after 24 h, a 46% decrease in TGF-β1 after 24 h, and a 54% and 59% decrease in PDGF-AB/BB after 24 h, respectively. This is consistent with a previous report by Curigliano et al., who found that breast cancer patients demonstrated a continuous decrease in perioperative plasma levels of VEGF, TGF-β1, and FGF2 until POD5 [77]. The same group surveyed three types of surgical procedures in these patients varying in the magnitude of tissue injury, including minimally invasive quadrantectomies, moderate mastectomies without reconstruction, and heavy mastectomy followed by reconstruction with trans- vs. recto-abdominal muscle cutaneous flap (TRAM). Interestingly, there was no significant difference in plasma VEGF, TGF-β1, and FGF2 levels during the perioperative period between these surgical groups, although the analysis was only conducted until POD5. Hormbrey et al. reported similar findings in breast cancer patients, noting that compared to preoperative values, plasma VEGF was significantly reduced over the first three days after surgery, and after that, levels recovered [78]. In colon cancer patients, serum VEGF levels [79] and PDGF-AB/BB [80,81] significantly dropped postoperatively. In renal cancer resection, Klatte et al. found that soon after surgery, peripheral venous VEGF, PDGF-AB/BB, and TGF-β1 levels were significantly decreased, whereas angiostatic molecules such as endostatin increased [82]. Perhaps most importantly, renal venous VEGF, PDGF-AB/BB, and TGF-β1 levels were higher than in the general venous blood pool, suggesting that the tumor-bearing kidney is the source of these factors. When this source is removed via nephrectomy, the levels decrease according to their baseline values at POD5. The report indicated that the higher renal venous levels may be a consequence of alterations in the hypoxia-induced pathway in renal cancer, associated with the loss of von Hippel–Lindau (VHL); accumulation of hypoxia-inducible factor (HIF)-1α; and consequent overexpression of VEGF, PDGF-AB/BB, and TGF-β1, which are then upregulated in renal venous blood [83]. Returning to pre-surgical levels implies that compensatory mechanisms are activated, similar to the later phases of the wound-healing response [82,84].

As previously mentioned, it has been suggested that the drop in plasma VEGF, TGF-β1, FGF2, and PDGF-AB/BB during the perioperative period may be attributed to the removal of the tumor and its associated stromal and endothelial cells that are responsible for generating these factors [79,82,84,85]. However, it is essential to note that the levels of these angiogenic molecules do not reflect their concentration in wound fluid at the resected tissue site [86,87]. For VEGF, it has been shown that VEGF levels in wound fluid are significantly higher compared to postoperative plasma or plasma levels. Wound fluid VEGF levels significantly increase immediately after surgery, a response directly linked to tissue injury. Thus, surgery-associated VEGF regulation may be more profoundly studied in locally generated wound fluid than in blood [86]. This is consistent with other reports showing that angiogenic factors in the wound environment increase postoperatively and are robustly higher than in plasma, where their concomitant decrease is observed [78]. For FGF2, it has been shown that the action of heparin sulfate-degrading enzymes during tissue injury can activate FGF2 locally and stimulate its production in wound fluid to enhance angiogenesis. Thus, the acute wound-healing response may act as a “molecular trap” for angiogenic factors, thereby reducing their plasma/plasma levels as a direct result [77,78].

Furthermore, since platelets are recruited to the site of tissue injury during wound healing, the postoperative drop in platelet count may also explain the downregulation of PDGF-AB/BB in the plasma. The role of TGF-β1 is context-dependent; it is considered both a pro- and anti-oncogenic protein involved in promoting and inhibiting inflammation. It is an angiogenic factor associated with aggressive phenotypes and poor patient prognosis through its paracrine action in the tumor microenvironment. Similar to other angiogenic factors, its significant decline in the postoperative period can be attributed to removing the tumor and associated stromal and endothelial cells involved in producing these factors [88,89]. Although we have not analyzed wound fluid, the local and elevated production of these angiogenic growth factors in the resected area may represent an essential mechanism for initiating the local recurrence of residual cancer.

Evidence from rodent models and retrospective clinical reports support the hypothesis that surgical procedures induce systemic immunosuppression. The effects of surgical anesthesia and analgesics, the increased release of glucocorticoids and catecholamines, blood transfusions, and surgery-associated mild hypothermia contribute to perioperative immunosuppression [90,91,92]. The perioperative period is also characterized by reduced numbers and activities of NK cells, Th1 cells, and cytotoxic T lymphocytes (CTL) involved in cell-mediated immunity and immunosurveillance of tumor cells [91]. Indeed, there is an increased frequency of metastases observed in immunocompromised patients, including those receiving immunosuppressant therapy [93,94,95]. Our findings in the present report indicated that decreased plasma levels of distinct pro-inflammatory cytokines and chemokines may mediate this perioperative immunosuppression. We observed a significant decrease in plasma IL-1α and leukemia inhibitory factor (LIF) and a similar but non-significant decrease in IL-1β and TNF-α 24 h after surgery in comparison to preoperative levels. These findings are consistent with Neagu et al.’s report, which analyzed the perioperative plasma profile of colorectal cancer patients vs. healthy controls [64]. They found that although the baseline plasma levels of IL-1 and TNF-α were elevated in cancer patients concerning healthy controls, plasma levels decreased postoperatively in all subjects and normalized within one week.

We also observed a decrease in the circulating levels of MCP-2 (CCL8) and SDF-1α/β (CXCL12) chemokines. Normally, MCP-2 activates mast cells, eosinophils, basophils, monocytes, T cells, and NK cells, many of which have potent anti-tumor and immunosurveillance activity [61]. A protective role for MCP-2 has previously been reported in a mouse model of B16F10 [96] and B78/H1 melanoma [97], where MCP-2 was found to inhibit cancer cell migration and reduce metastasis. Specifically, MCP-2 production was found to be regulated by STAT3, and STAT3-induced MCP-2 production was concomitant with a reduction in IL-6 and TNF-α levels in vivo [96]. Thus, it was suggested that hyperactivation of STAT3 in myeloid cells simultaneously exerted anti-inflammatory and anti-tumor effects via MCP-2 induction [96].

In contrast, upregulated levels of angiogenic chemokine SDF-1 and subsequent binding to CXCR4, the cognate receptor for SDF-1, can induce angiogenesis and promote tumor progression [98,99]. Elevated SDF-1 levels have rapidly recruited bone marrow-derived progenitors and CXCR4-expressing cancer cells to the pre-metastatic niche [100]. Studies blocking the vital chemotactic function of SDF-1 using CXCR4 inhibitor AMD-3100 reported increased effectiveness of a vascular-disrupting agent, combretastatin, in a mouse model of breast cancer, presumably by preventing TIE2-expressing macrophages from being recruited to tumors [101]. The postoperative decrease in plasma MCP-2 and SDF-1 chemokines may indicate reduced secretion as lymphocytes are mobilized to wounded tissue postoperatively. Thus, it may be beneficial to analyze the expression profile of these chemokines in wound fluid draining the resected tissue site. There were no discernable changes in plasma MCP-1 (CCL2) or IL-8 (CXCL8) chemokines during the perioperative period. This may be because activated platelets locally secrete both MCP-1 and IL-8 during wound healing to initiate the recruitment of various leukocytes. Both MCP-1 and IL-8 have been implicated in tumor progression by suppressing the tumor-infiltration and anti-tumor activity of monocytes [102] and stimulating angiogenesis and PLC-β and PI3K signaling in endothelial cells [61], respectively. Although a moderate increase is noted for both MCP-1 and IL-8 after surgery, consistent with other reports [64,103,104,105,106,107], their localized secretion and chemotactic activity during wound healing suggests that they may be more accurately profiled in wound fluid. The plasma profile we observed after surgery suggests a marked immunosuppressive response. Given the increasingly important role played by immune checkpoint blockers, designed to activate the immune response against cancer in treating cancer in the adjuvant setting, this is potentially of great clinical importance.

The perioperative plasma profile also reflects the acute phase inflammatory response following surgical tissue injury. The magnitude of this response and its consequent production of pro-inflammatory mediators, such as IL-6, has been directly linked to the invasiveness of the procedure [55,56,57,59,108]. In colorectal cancer, the minimally invasive laparoscopic technique is associated with significantly lower levels of plasma IL-6 after surgery, reduced local inflammation, and better overall outcomes compared to the open procedure [108]. These clinical reports indicate that all surgical interventions, although varying in degrees, are followed by a sustained release of IL-6 that can be detected within one hour after surgery. Indeed, we observed a significant increase in IL-6 postoperatively, consistent between all cancer types and reaching peak levels on POD1 before normalizing to baseline. Plasma levels of IL-6 directly indicate its availability in circulation and peripheral tissues, signifying its potential to stimulate CTCs and distant micrometastases. The mitogenic roles of IL-6 are well established. IL-6 binding to cancer cells is capable of initiating several mitogenic signaling cascades via activation of the PI3K/Akt, ERK/MAPK, and the JNK/STAT3 pathways, resulting in increased cancer cell survival, proliferation, malignant transformation and dedifferentiation, migration, and invasion [109,110,111,112,113,114]. IL-6 has also been shown to promote epithelial–mesenchymal transition (EMT) in several cancers through the altered expression of E- and N-cadherin [111,115,116], Vimentin [117], Snail [109,116,118], Twist [119,120], and others [109]. The expression of epithelial marker E-cadherin is repressed. In contrast, the increased expression of mesenchymal markers such as N-cadherin and Vimentin and transcription factors Snail and Twist are linked to the induction of EMT. The mesenchymal phenotype enhances cells’ ability to invade, migrate, escape apoptosis, and produce extracellular matrix. The epithelial cell is characterized by polarity, adherence to the basement membrane, and other epithelial cells via tight cellular junctions involving E-cadherin. In contrast, the increased expression of N-cadherin, cytoskeletal protein Vimentin, and matrix metalloproteinases enable invasion and mobility. This process comes into play during wound healing to fill the breach and restore epidermal continuity. Its involvement in pathology corresponds to the local invasion of cancer cells, enabling metastatic progression of the cancer. This is a reversible phenotype with intermediate states. The mesenchymal–epithelial transition (MET) is involved in reimplanting distant metastases after migration [121]. Upregulation of IL-6 in plasma levels of mice has been shown to promote tumor growth, angiogenesis, metastasis, and the expansion of cancer stem cells, as well contribute to multidrug resistance via gp130/MAPK/STAT3-mediated activation of C/EBPβ/δ transcription factors [109,122]. Thus, due to the extensive tumorigenic role of IL-6, its rapid surge postoperatively has critical implications for triggering residual disease progression.

Perhaps the most noteworthy change observed during the perioperative period was the plasma profile of hepatocyte growth factor (HGF), a ligand known for its role in tumorigenesis and malignant progression [110]. Almost all patients demonstrated a rapid increase in plasma HGF, reaching peak levels of POD1 in breast and colorectal cancer patients and POD3 in prostate cancer patients, consistent with other reports [123,124,125,126]. Notably, colorectal cancer patients exhibited the most significant upregulation in plasma HGF—approximately a 6-fold increase within 24 h after surgery. For patients harboring residual disease or micrometastases, the transient yet drastic upregulation of HGF may have significant implications. Elevated HGF levels can stimulate the growth and proliferation of CTCs, epithelial cells, fibroblasts, and other stromal cells that promote tumor growth, invasion, metastasis, and angiogenesis [127,128,129,130]. HGF-dependent MET activation has also been linked to drug resistance against EGFR tyrosine kinase inhibitors, ALK inhibitors, and RAF inhibitors [131,132,133,134,135,136,137]. Indeed, upregulation in plasma and plasma HGF, soluble MET, and phosphor-MET have been associated with primary and distant tumor progression, resistance to therapy, and overall poorer outcomes.

Unlike HGF, other growth factors analyzed did not demonstrate a similar plasma profile during the perioperative period. We observed a decrease in plasma IGF-1 and EGF after surgery, varying among cancer types. Consistent with these findings, Kumara et al. found that the postoperative rise in HGF was also accompanied by a decrease in plasma EGF and IGF-1 in patients undergoing minimally invasive colorectal tumor resection [51]. This decrease was partly attributed to removing the primary tumor and its associated endothelial and stromal cells that produce these factors. The same study reported a postoperative rise in PLGF that persisted for three weeks [51]. PLGF, a member of the VEGF subfamily, is well known for its role in promoting tumor growth, neovascularization, and metastasis, and high levels of PLGF have become a prognostic indicator of disease recurrence [52,53]. We also observed a substantial increase in plasma PLGF in prostate cancer patients, yet only within the first 48 h following surgery.

In our findings, inter-individual variability can be seen in all cancer types. Many factors can account for some individuals having slightly higher inflammatory cytokines, such as lifestyle (high-inflammatory diets and exercise) and/or the presence of possible infections [138,139]. However, since we were tracking the same patients throughout the study, this variability does not impact the overall findings.

The variability post-surgery may be a result of the surgical technique used, as inflammation and the extent of tissue damage are often linked. However, there is no consensus on this topic, as multiple studies have found contradicting results. As previously mentioned, one study found no difference in angiogenic cytokine levels after minimally invasive quadrantectomies compared to moderate mastectomies without reconstruction and heavy mastectomy in breast cancer patients [77]. A multicenter randomized controlled trial analyzed the inflammatory cytokine levels of 38 patients who had undergone either laparoscopic (LPD) or open pancreatoduodenectomy (OPD). In this study, the levels of several inflammatory markers, namely IL-6, TNF-α, IL-1β, IL-8, and C reactive protein (CRP), were measured up to 96 h after incision to ascertain whether minimally invasive surgery was correlated with lower systemic inflammatory responses [140]. The authors found that there were no significant differences in the extent of the inflammatory response between the two surgical types for pancreatoduodenectomy. Contrastingly, in another study that looked at laparoscopic versus conventional open appendectomy, postoperative IL-6 levels were significantly lower in the former group, signifying that there may be a potential correlation between the magnitude of surgical stress and inflammatory markers [141]. In colonic cancer patients, laparoscopic surgery conferred lower serum IL-6 and VEGF levels compared to open colectomy; however, local levels of the inflammatory and angiogenic cytokines were not significantly different between the two groups [142]. These findings suggest that while the surgical technique may potentially impact the cytokine levels, there is no standardized basis, and variety between surgical groups should be expected.

## 5. Conclusions

Here, we provide evidence to show that significant changes in cancer patients’ immunologic and angiogenic plasma cytokine profiles characterize the perioperative period following tumor resection. Immediate increases in MMP-9, IL-6, and HGF paired with decreases in angiogenic growth factors such as VEGF, FGF2, TGF-β1, PDGF-AB/BB, and growth factors like IGF-1 and EGF are noteworthy and illustrated in Figure 20. This systemic disturbance explains how surgical intervention may stimulate local or metastatic tumor growth and facilitate metastatic colonization of circulating tumor cells (CTCs). While most of these cytokines return to baseline levels within 48 h to a couple of weeks, this brief inflammatory state may be sufficient in triggering any residual cells to undergo EMT. An unproven hypothesis is that minimizing local inflammation and the consequent systemic immunosuppression of open surgery can potentially reduce the risk of disease recurrence. These findings may also underscore the importance of initiating adjuvant therapy during the perioperative period and the risks of delaying treatment [143,144,145]. There is the potential for this research to be extended to other types of cancer, such as the highly aggressive pancreatic cancer. Insight into the effects of primary tumor removal and surgical wound healing on the behavior of residual disease may uncover a better approach to the timing, duration, and design of perioperative or adjuvant therapies.

## 6. Limitations

There is a multitude of metabolic changes that occur throughout the body after surgery as a result of the surgical technique, anesthesia, and nutrition. These changes particularly impact the hypothalamic–pituitary axis, resulting in increases in the stress hormone cortisol and insulin-like growth factors (IGFs), glucose utilization impairment, and dysregulation of thyroid hormone production [147]. These changes are highly complex and often contrast each other. For example, many colorectal cancer patients are considered malnourished before surgery, which has been shown to impair IGF-1 function, leading to chronic inflammation [148,149]. Surgery-induced upregulation of IGF-1 can further increase inflammation, potentially accounting for the high levels of inflammatory markers seen in this cancer subtype. In contrast, high levels of cortisol are linked to the inhibition of inflammatory cytokine production and glucocorticoid resistance [150]. For this reason, the levels of inflammatory cytokines immediately after surgery may be reduced due to high serum cortisol. As our current study did not investigate these changes, we acknowledge that they may have an important impact on these findings, highlighting an avenue for future studies.

In a subset of patients, postoperative tumor recurrence and metastasis occurred, ultimately leading to patient death. Data pertaining to other patients are unavailable as they did not follow up after tumor resection. Nevertheless, our investigations, coupled with current understandings of the interplay of cytokines and tumor metastasis, suggest that it is likely for tumor recurrence to occur [151,152].

## Figures and Tables

**Figure 1 cells-12-02767-f001:**
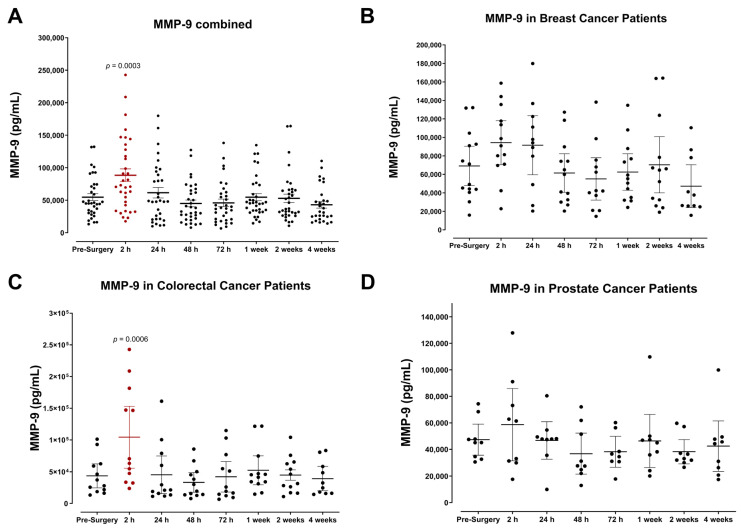
Serum matrix metalloproteinase-9 (MMP-9) levels in surgical cancer patients during the perioperative period. (**A**) Combined patient analysis indicates a significant increase in plasma MMP-9 two hours after surgery compared to pre-surgery levels (*p* = 0.0003, n = 35). MMP-9 levels quickly returned to baseline within 24–48 h after surgery. Although not significant, there also appears to be a postoperative increase in MMP-9 plasma levels in (**B**) breast (n = 14) and (**D**) prostate cancer patients (n = 9). (**C**) Colorectal patients exhibited a significant increase in plasma MMP-9 two hours after surgery compared to pre-surgery levels (*p* = 0.0006, n = 12). All patient plasma samples were assayed in duplicate, and results were analyzed and compared using one-way analysis of variance (ANOVA) at 95% confidence using Fisher’s LSD test.

**Figure 2 cells-12-02767-f002:**
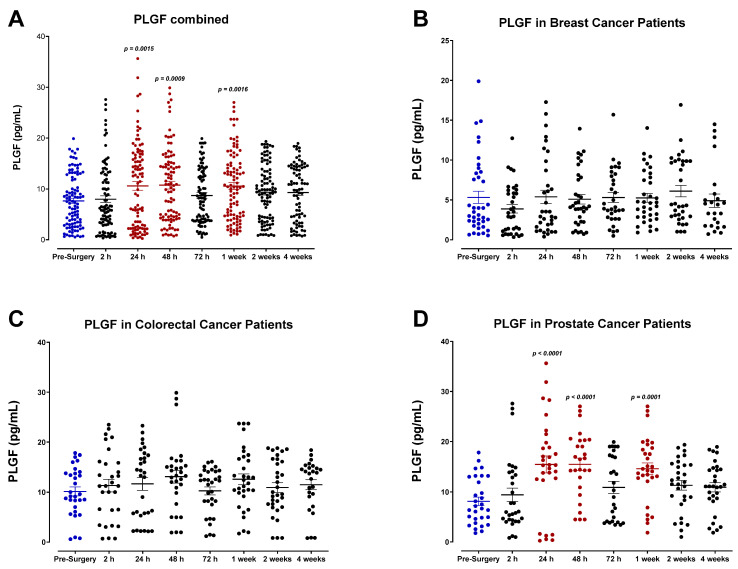
Plasma placental growth factor (PLGF) levels in surgical cancer patients during the perioperative period. (**A**) Combined patient analysis (n = 36) indicates a significant increase in plasma PLGF 24 h (*p* = 0.0015), 48 h (*p* = 0.009), and 1 week (*p* = 0.0016) after surgery in comparison to pre-surgery levels. (**B**) Breast cancer patients (n = 14) demonstrated attenuated overall PLGF levels, while colorectal cancer patients (n = 12) (**C**) showed no discernable trend compared to pre-surgery. In contrast, (**D**) prostate cancer patients exhibited a significant increase in PLGF levels at 24 h, 48 h, and 1 week (*p* < 0.0001, n = 10) after surgery in comparison to pre-surgery levels. All patient plasma samples were assayed in duplicate, and results were analyzed and compared using one-way analysis of variance (ANOVA) at 95% confidence using Fisher’s LSD test.

**Figure 3 cells-12-02767-f003:**
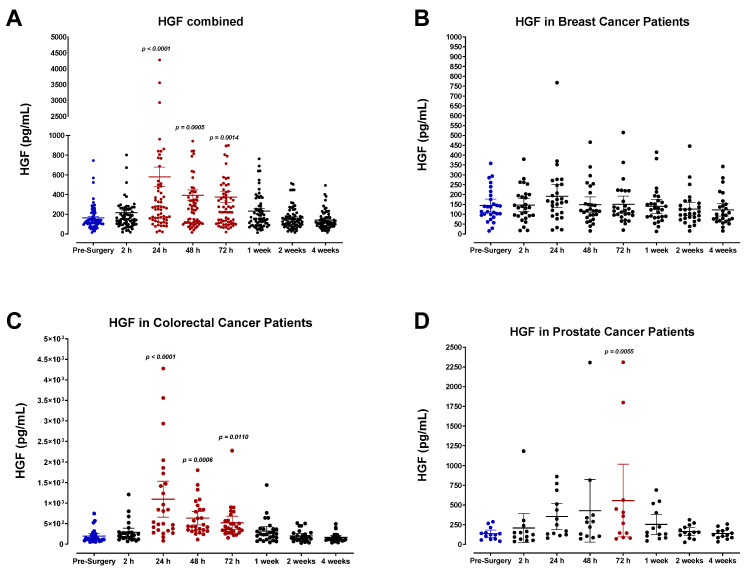
Plasma hepatocyte growth factor (HGF) levels in surgical cancer patients during the perioperative period. (**A**) Combined patient analysis indicates a significant increase in plasma HGF levels 24 h after surgery compared to pre-surgery levels (*p* < 0.0001, n = 68). HGF levels remained significantly elevated 48 h (*p* = 0.0005) and 72 h (*p* = 0.0014) after surgery. HGF levels returned to baseline approximately one week after surgery. (**B**) Breast cancer patients (n = 28) showed a non-significant increase in plasma HGF levels, while (**C**) colorectal cancer patients (n = 27) exhibited higher levels of plasma HGF after 24 h (*p* < 0.0001). The trend was sustained until 72 h post-surgery (*p* = 0.0006, 48 h; *p* = 0.0110, 72 h). (**D**) Prostate cancer patients (n = 13) demonstrated a delayed trend with an increase at 72 h (*p* = 0.0055) before returning to baseline. All patient plasma samples were assayed in duplicate, and results were analyzed and compared using one-way analysis of variance (ANOVA) at 95% confidence using Fisher’s LSD test.

**Figure 4 cells-12-02767-f004:**
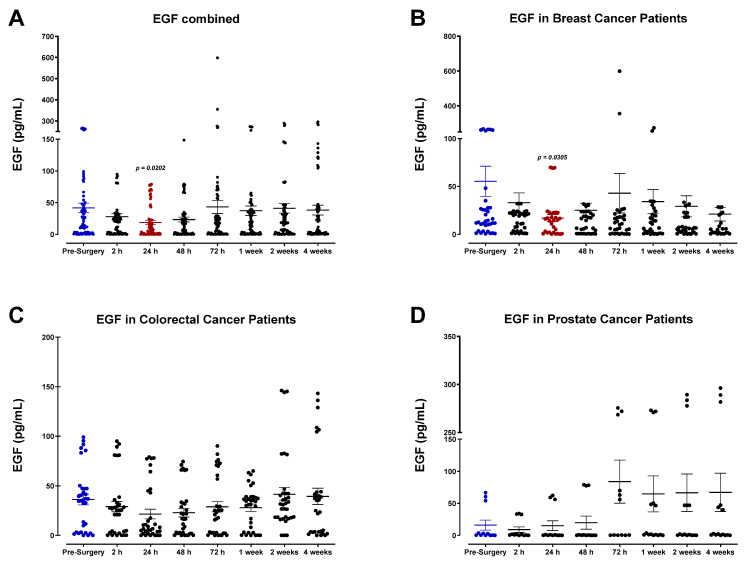
Plasma epidermal growth factor (EGF) levels in surgical cancer patients during the perioperative period. (**A**) Combined patient analysis (n = 29) and (**B**) breast cancer patient (n = 13) data indicate a noteworthy but non-significant decrease in plasma EGF levels approximately 24 h after surgery (*p* = 0.0202 and 0.0305, respectively). EGF levels appeared to return to baseline approximately 72 h after surgery. (**C**) Colorectal (n = 11) and (**D**) prostate cancer patients (n = 5) had no significant increases compared to pre-surgery levels. All patient plasma samples were assayed in duplicate, and results were analyzed and compared using one-way analysis of variance (ANOVA) at 95% confidence using Fisher’s LSD test.

**Figure 5 cells-12-02767-f005:**
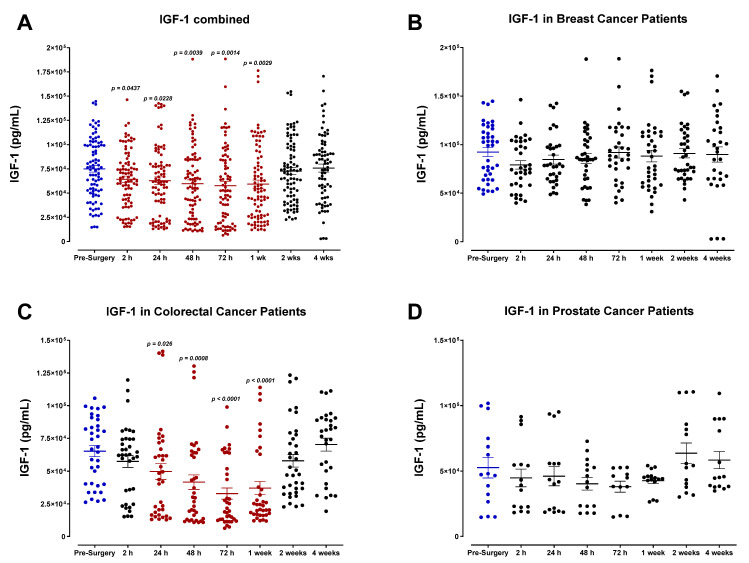
Plasma insulin-like growth factor-I (IGF-1) levels in surgical cancer patients during the perioperative period. (**A**) Combined patient analysis indicates a significant decrease in plasma IGF-1 as early as 2 h after surgery compared to pre-surgery levels (n = 30, *p* = 0.0437). IGF-1 levels were maintained and returned to baseline 2–4 weeks after surgery. Despite an apparent postoperative decrease, there was no significant trend in plasma levels of IGF-1 noted in (**B**) breast cancer patients (n = 13) or (**D**) prostate cancer patients (n = 5). Among each cancer type, (**C**) colorectal cancer patients demonstrated the most significant postoperative decrease in plasma IGF-1, particularly 72 h after surgery (n = 12, *p* < 0.0001). All patient plasma samples were assayed in duplicate, and results were analyzed and compared using one-way analysis of variance (ANOVA) at 95% confidence using Fisher’s LSD test.

**Figure 6 cells-12-02767-f006:**
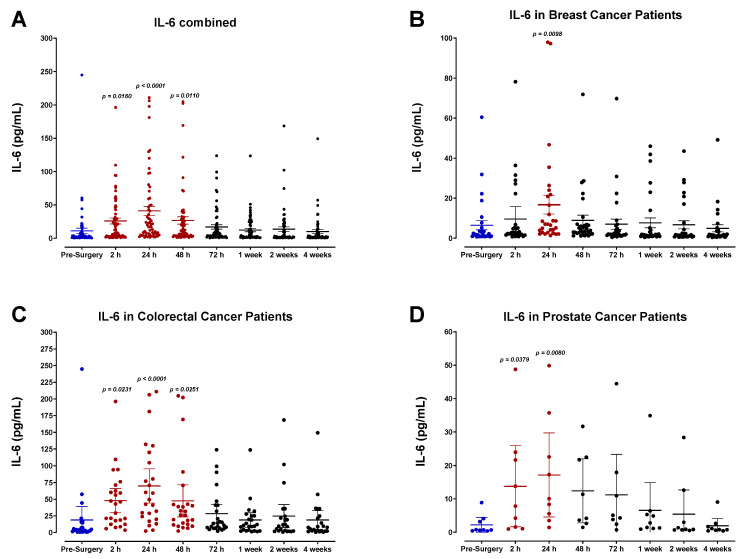
Plasma levels of IL-6 in surgical cancer patients during the perioperative period. (**A**) Combined patient analysis indicates a significant increase in plasma IL-6 approximately 2 h after surgery, reaching its peak at 24 h compared to pre-surgery levels (*p* < 0.000, n = 66). After this time, IL-6 levels steadily returned to baseline within 72 h after surgery. (**C**) Colorectal cancer patients (*p* < 0.0001, n = 25) demonstrated significantly higher overall IL-6 plasma levels in comparison to (**B**) breast (*p* = 0.0098, n = 31) and (**D**) prostate cancer patients (*p* = 0.008, n = 10). All patient plasma samples were assayed in duplicate, and results were analyzed and compared using one-way analysis of variance (ANOVA) at 95% confidence using Fisher’s LSD test.

**Figure 7 cells-12-02767-f007:**
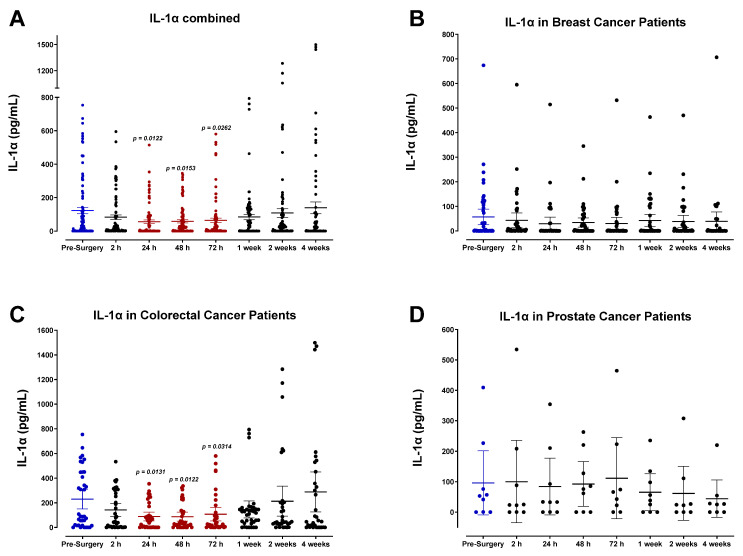
Plasma levels of IL-1α in surgical cancer patients during the perioperative period. (**A**) Combined patient analysis indicates a trending significant decrease in plasma IL-1α approximately 24–72 h after surgery in comparison to pre-surgery levels (*p* < 0.0262, n = 53). After this time, IL-1α levels steadily returned to baseline within four weeks after surgery. (**C**) Colorectal cancer patients (n = 22) demonstrated the most significant postoperative decrease in plasma IL-1α, particularly 24 h (*p* = 0.0131) and 48 h (*p* = 0.0122) after surgery. Despite an apparent postoperative decrease in the combined plasma analysis, there was no significant trend in plasma levels of IL-1α noted in (**B**) breast cancer patients (n = 28) or (**D**) prostate cancer patients (n = 3). All patient plasma samples were assayed in duplicate, and results were analyzed and compared using one-way analysis of variance (ANOVA) at 95% confidence using Fisher’s LSD test.

**Figure 8 cells-12-02767-f008:**
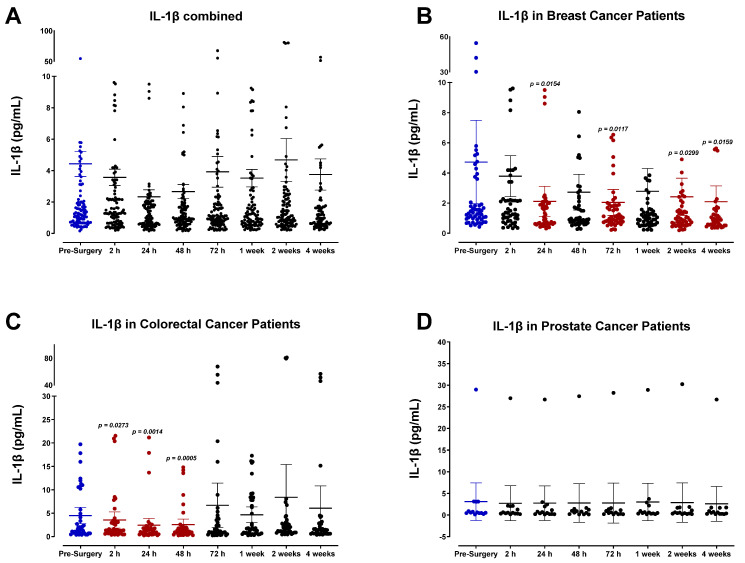
Plasma levels of IL-1β in surgical cancer patients during the perioperative period. (**A**) Combined patient analysis indicates a trending but non-significant decrease in plasma IL-1β approximately 24–48 h after surgery in comparison to pre-surgery levels (n = 65). After this time, IL-1β levels returned to baseline approximately two weeks after surgery. Overall, plasma analysis of IL-1β levels demonstrated variation among individual patients and cancer types. (**B**) Breast cancer patients (n = 30) had fluctuation, with a significant decrease at 24 h (*p* = 0.0154), then increased back to baseline before decreasing again at 72 h (*p* = 0.0117), 2 weeks (*p* = 0.0299), and 4 weeks (*p* = 0.0159). IL-1β plasma levels in (**C**) colorectal cancer patients (n = 25) decreased as early as 2 h and then returned to baseline after 72 h, with the most significant decrease seen at 24 h (*p* = 0.0014). No difference was seen in (**D**) prostate cancer patients (n = 10). All patient plasma samples were assayed in duplicate, and results were analyzed and compared using one-way analysis of variance (ANOVA) at 95% confidence using Fisher’s LSD test.

**Figure 9 cells-12-02767-f009:**
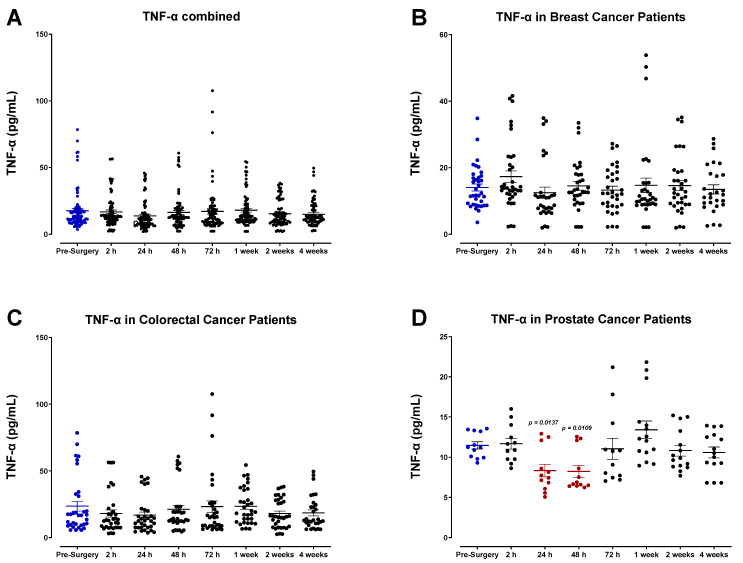
Plasma tumor necrosis factor (TNF-α) levels in surgical cancer patients during the perioperative period. (**A**) Combined patient analysis indicates high variation among each timepoint analyzed. There is also a trending but non-significant decrease in plasma TNF-α approximately 24 h after surgery compared to pre-surgery levels (n = 29). (**B**) Despite an apparent postoperative decrease in the combined plasma analysis, there was no significant trend in plasma levels of TNF-α noted in breast (n = 13) or (**C**) colorectal cancer patients (n = 11). However, there was a significant postoperative decrease in plasma TNF-α observed in (**D**) prostate cancer patients (*p* < 0.0137, n = 5) approximately 24–48 h after surgery before returning to baseline levels. All patient plasma samples were assayed in duplicate, and results were analyzed and compared using one-way analysis of variance (ANOVA) at 95% confidence using Fisher’s LSD test.

**Figure 10 cells-12-02767-f010:**
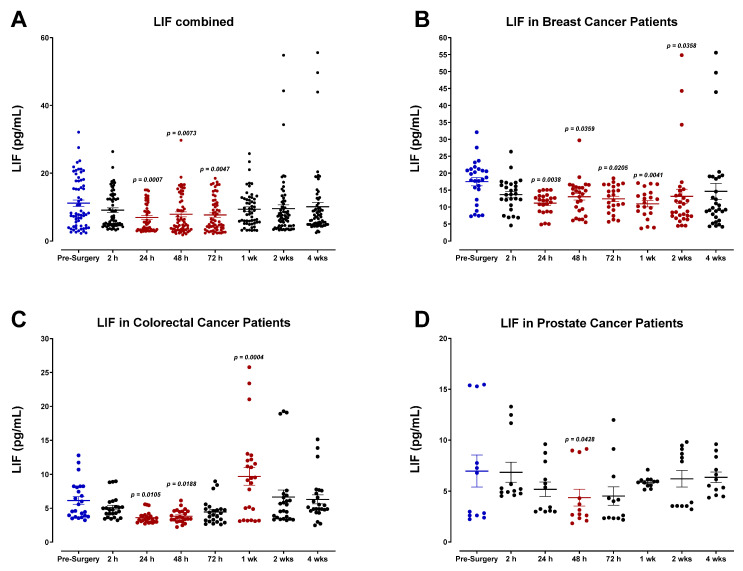
Plasma levels of leukemia inhibitory factor (LIF) in surgical cancer patients during the perioperative period. (**A**) Combined patient analysis indicates a significant decrease in plasma LIF approximately 24 h after surgery compared to pre-surgery levels (*p* = 0.0007, n = 19). After this time, LIF levels steadily returned to baseline within one week after surgery. (**B**) Breast cancer patients demonstrated significantly higher overall LIF plasma levels in comparison to (**C**) colorectal (*p* = 0.0105, n = 7) and (**D**) prostate cancer patients (*p* = 0.0428, n = 3). Interestingly, LIF levels in colorectal cancer patients increase above baseline at approximately one week post-surgery before decreasing again. All three cancer types return to baseline levels around two weeks post-surgery. All patient plasma samples were assayed in duplicate, and results were analyzed and compared using one-way analysis of variance (ANOVA) at 95% confidence using Fisher’s LSD test.

**Figure 11 cells-12-02767-f011:**
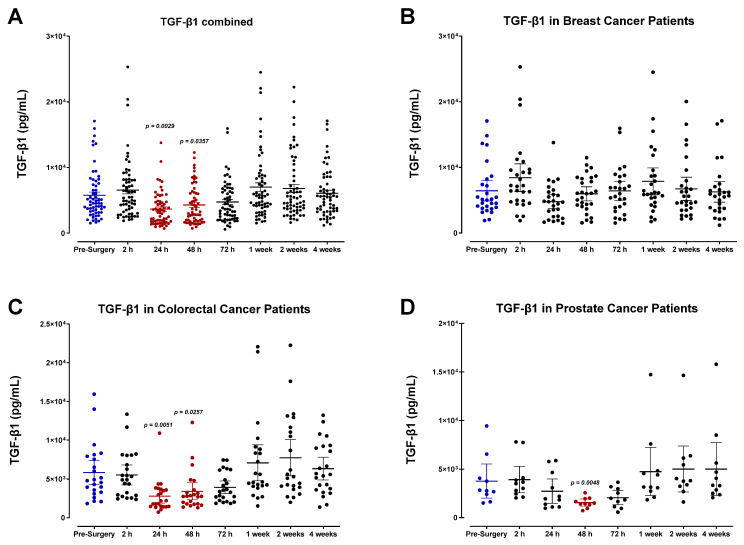
Plasma transforming growth factor (TGF-β1) levels in surgical cancer patients during the perioperative period. (**A**) Combined patient analysis indicates a significant decrease in plasma TGF-β1 measured 24 and 48 h after surgery (*p* < 0.0357, n = 29). TGF-β1 levels quickly returned to baseline within one week after surgery. (**B**) In breast cancer patients (n = 13), there was no significant difference in plasma TGF-β1 measured between pre- and post-surgery. In contrast, (**C**) colorectal cancer patients (n = 12) had a significant decrease at 24 h (*p* = 0.0051) and 48 h (*p* = 0.0257) compared to pre-surgery levels. Similarly, (**D**) in prostate cancer patients (n = 4), there was a significant decrease in plasma TGF-β1 levels measured after 48 h (*p* = 0.0048). All patient plasma samples were assayed in duplicate, and results were analyzed and compared using one-way analysis of variance (ANOVA) at 95% confidence using Fisher’s LSD test.

**Figure 12 cells-12-02767-f012:**
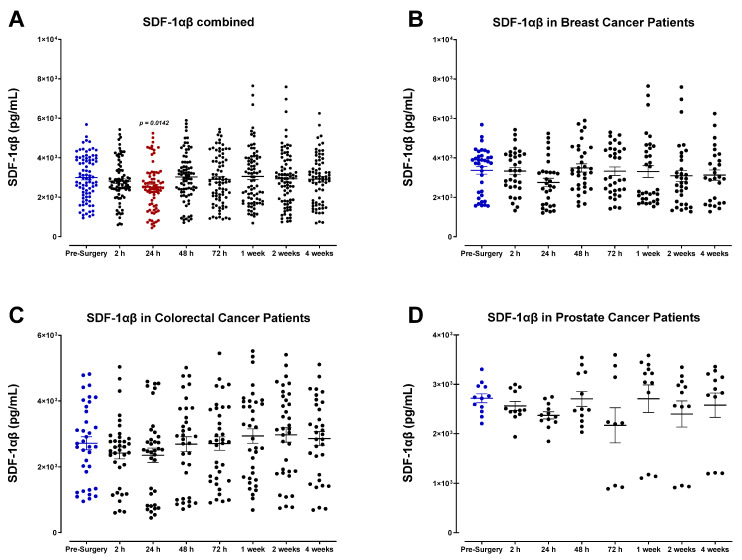
Plasma levels of stromal cell-derived factor (SDF-1) α/β in surgical cancer patients during the perioperative period. (**A**) Combined patient analysis indicates a slightly significant drop in plasma SDF-1α/β approximately 24 h after surgery compared to pre-surgery levels (*p* = 0.0142, n = 29). SDF-1α/β levels quickly returned to baseline within 48 h after surgery. There was no significant difference in SDF-1α/β plasma levels observed during the perioperative period in individual cancer types: (**B**) breast (n = 13), (**C**) colorectal (n = 12), and (**D**) prostate cancer patients (n = 4). All patient plasma samples were assayed in duplicate, and results were analyzed and compared using one-way analysis of variance (ANOVA) at 95% confidence using Fisher’s LSD test.

**Figure 13 cells-12-02767-f013:**
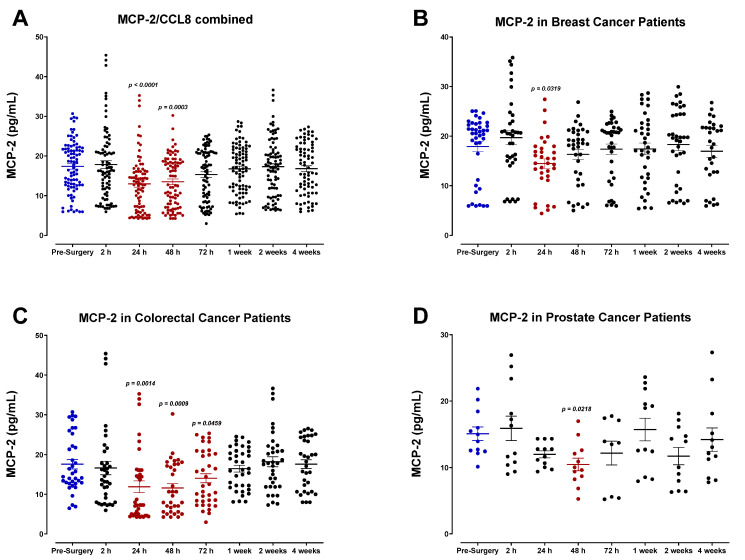
Plasma monocyte chemoattractant protein (MCP-2) levels in surgical cancer patients during the perioperative period. (**A**) Combined patient analysis indicates a significant decrease in plasma MCP-2 24 h (*p* < 0.0001) and 48 h (*p* = 0.0003) after surgery in comparison to pre-surgery levels (n = 29). MCP-2 levels quickly returned to baseline within 1–2 weeks after surgery. (**B**) Breast cancer patients (n = 13) also demonstrated a decrease in MCP-2 levels 24 h after surgery (*p* = 0.0319) prior to returning to baseline, whereas (**C**) colorectal cancer patients (n = 12) exhibited a maintained drop from 24–72 h (*p* < 0.0459). (**D**) Prostate cancer patients (n = 4) had fluctuating levels when compared to pre-surgery levels; however, they were only significant at 48 h (*p* = 0.0218). All patient plasma samples were assayed in duplicate, and results were analyzed and compared using one-way analysis of variance (ANOVA) at 95% confidence using Fisher’s LSD test.

**Figure 14 cells-12-02767-f014:**
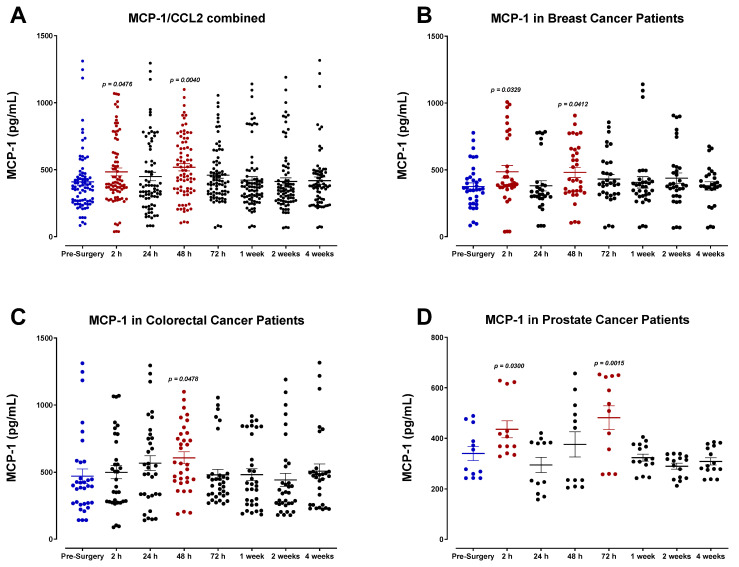
Plasma monocyte chemoattractant protein MCP-1 levels in surgical cancer patients during the perioperative period. (**A**) Combined patient analysis indicates increases in plasma MCP-1 levels at 2 h (*p* = 0.0476) and 48 h (*p* = 0.0040) post-surgery (n = 29). A similar trend is seen in (**B**) breast cancer patients (*p* < 0.0412, n = 13), while (**D**) prostate cancer patients (n = 5) had increases at 2 h (*p* = 0.0300) and 72 h (*p* = 0.0015). (**C**) Colorectal cancer patients (n = 11) had a slight increase in plasma at 24 h; however, it was only significant at 48 h (*p* = 0.0478). All patient plasma samples were assayed in duplicate, and results were analyzed and compared using one-way analysis of variance (ANOVA) at 95% confidence using Fisher’s LSD test.

**Figure 15 cells-12-02767-f015:**
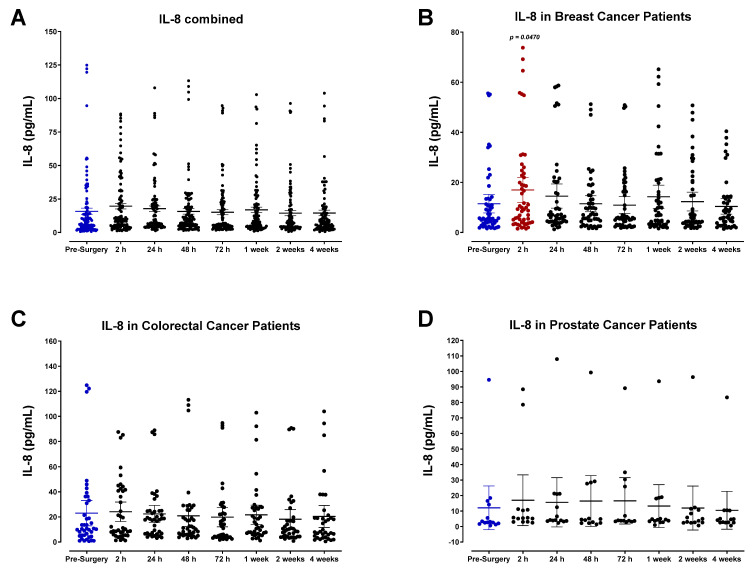
Plasma levels of IL-8 (CXCL8) in surgical cancer patients during the perioperative period. (**A**) Combined patient analysis indicates no discernable trend in plasma IL-8 levels during the perioperative period, and high variation is apparent among individual patients and cancer types (n = 65). (**B**) There is a significant increase in breast cancer patients at 2 h (*p* = 0.0470, n = 30) before returning to baseline levels after 48 h. There is no significant difference in IL-8 plasma levels observed during the perioperative period in (**C**) colorectal (n = 25) or (**D**) prostate cancer patients (n = 10). All patient plasma samples were assayed in duplicate, and results were analyzed and compared using one-way analysis of variance (ANOVA) at 95% confidence using Fisher’s LSD test.

**Figure 16 cells-12-02767-f016:**
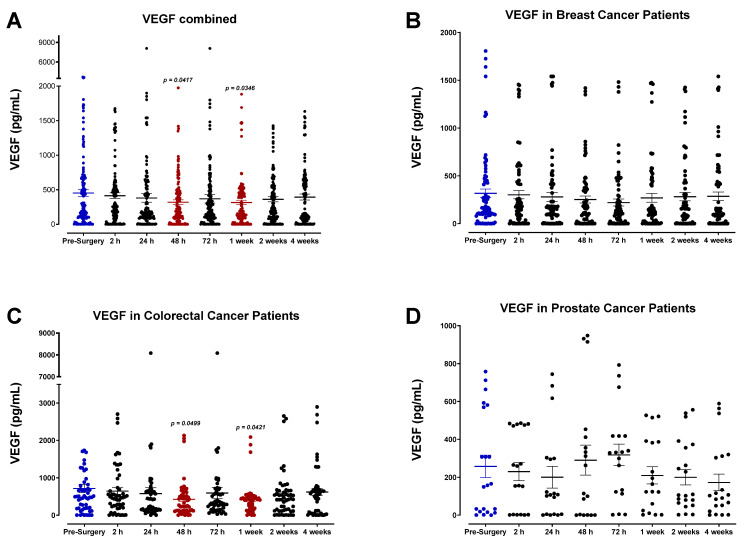
Plasma levels of vascular endothelial growth factor (VEGF) in surgical cancer patients during the perioperative period. (**A**) Combined patient analysis indicates a decreasing trend in postoperative plasma VEGF levels at 48 h (*p* = 0.0417) and 1 week (*p* = 0.0346) post-surgery (n = 56). Similarly, (**C**) colorectal cancer patients (n = 20) had significant decreases at 48 h (*p* = 0.0499) and 1 week (*p* = 0.0421). Despite a decreasing trend in postoperative plasma levels of VEGF, there is no significant difference in plasma levels observed during the perioperative period in (**B**) breast (n = 27) and (**D**) prostate cancer patients (n = 9). All patient plasma samples were assayed in duplicate, and results were analyzed and compared using one-way analysis of variance (ANOVA) at 95% confidence using Fisher’s LSD test.

**Figure 17 cells-12-02767-f017:**
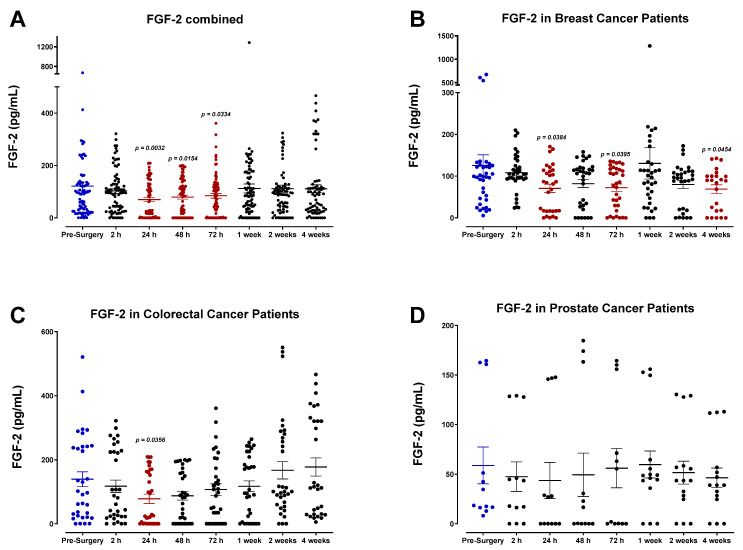
Plasma fibroblast growth factor (FGF-2) levels in surgical cancer patients during the perioperative period. (**A**) Combined patient analysis indicates a decreasing trend in plasma FGF-2 levels approximately 24 h after surgery (*p* = 0.0032), quickly returning to baseline 1-week post-surgery (n = 29). (**B**) Breast cancer patients (n = 13) had fluctuations in FGF-2 levels, with significant drops noted at 24 h (*p* = 0.0384), 72 h (*p* = 0.0395), and 4 weeks (*p* = 0.0454) post-surgery. (**C**) Colorectal cancer patients (n = 11) had a decrease at 24 h (*p* = 0.0356) before returning to baseline after 72 h. (**D**) prostate cancer patients (n = 5) had no significant changes in FGF-2 levels in the perioperative operative period. All patient plasma samples were assayed in duplicate, and results were analyzed and compared using one-way analysis of variance (ANOVA) at 95% confidence using Fisher’s LSD test.

**Figure 18 cells-12-02767-f018:**
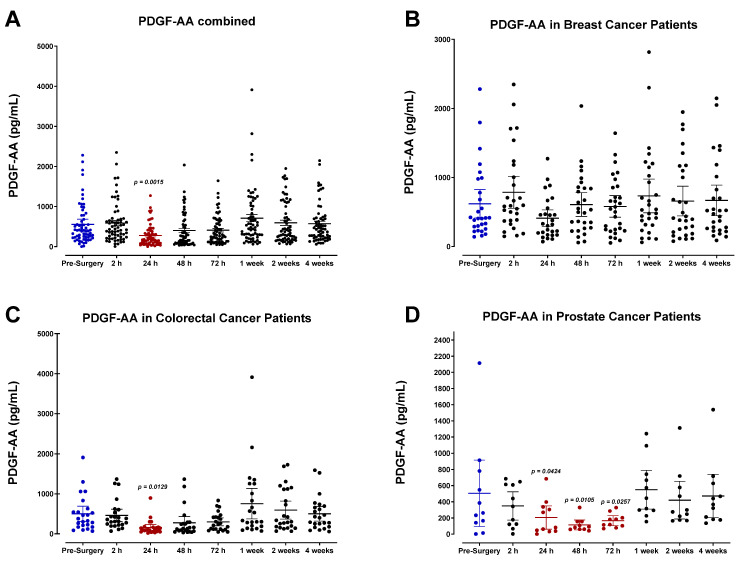
Plasma platelet-derived growth factor (PDGF-AA) levels in surgical cancer patients during the perioperative period. (**A**) Combined patient analysis indicates a significant decrease in plasma PDGF-AA measured approximately 24 h after surgery (*p* = 0.0015, n = 62) compared to pre-surgery levels. PDGF-AA levels quickly returned to baseline within one week after surgery. Despite a decreasing trend in postoperative plasma levels of PDGF-AA, no significant difference in plasma levels was observed during the perioperative period in (**B**) breast cancer patients (n = 28). (**C**) Colorectal cancer patients (n = 23) also had a decrease at 24 h (*p* = 0.0129) before returning to baseline levels, while (**D**) prostate cancer patients (n = 11) maintained decreased levels from 24 to 72 h (*p* < 0.0424) post-surgery. All patient plasma samples were assayed in duplicate, and results were analyzed and compared using one-way analysis of variance (ANOVA) at 95% confidence using Fisher’s LSD test.

**Figure 19 cells-12-02767-f019:**
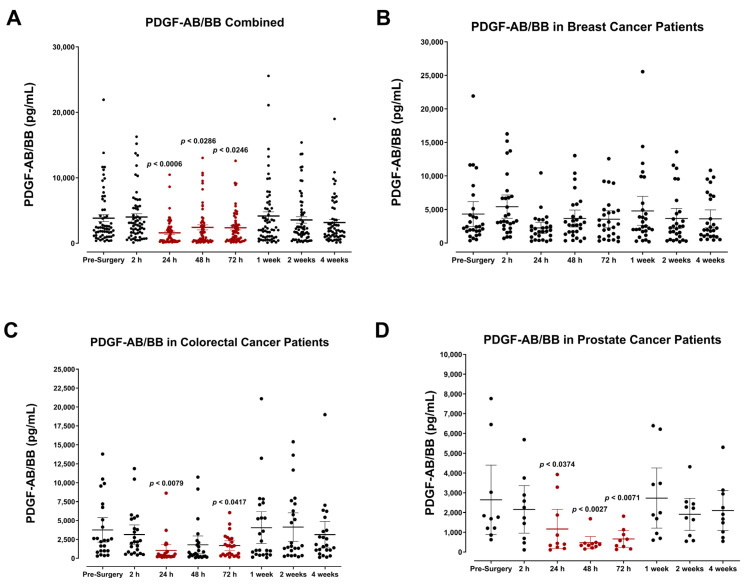
Plasma platelet-derived growth factor (PDGF-AB/BB) levels in surgical cancer patients during the perioperative period. (**A**) Combined patient analysis indicates a significant decrease in plasma PDGF-AB/BB measured approximately 24 h (*p* = 0.0006, n = 62), 48 h (*p* = 0.0286), and 72 h (*p* = 0.0246) after surgery. PDGF-AB/BB levels quickly returned to baseline within one week. Despite a decreasing trend in postoperative plasma levels of PDGF-BB, no significant difference in plasma levels was observed during the perioperative period in (**B**) breast cancer patients (n = 28). (**C**) Colorectal cancer patients (n = 23) had decreases noted in only the 24 h (*p* = 0.0079) and 72 h (*p* = 0.0417) post-surgery periods. (**D**) Prostate cancer patients (n = 11) similarly had significantly decreased levels of PDGF-AB/BB from 24 to 72 h post-surgery (*p* < 0.0374) before returning to preoperative levels. All patient plasma samples were assayed in duplicate, and results were analyzed and compared using one-way analysis of variance (ANOVA) at 95% confidence using Fisher’s LSD test.

**Figure 20 cells-12-02767-f020:**
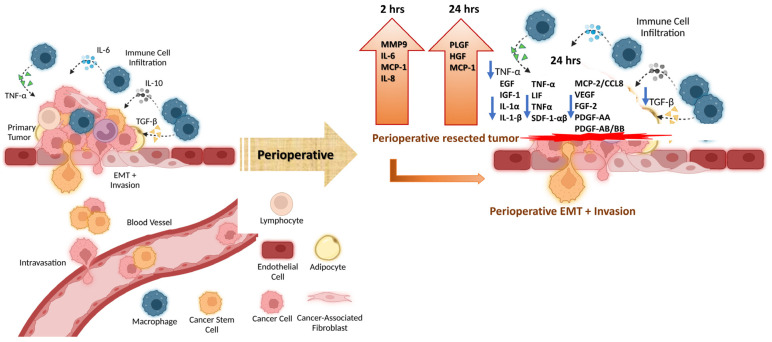
The paradox of pro-inflammatory and angiogenetic cytokines in host response against perioperative surgical breast, colorectal, and prostate cancer patients, demonstrating the complex fluctuation of cytokine profiling. Citation: Taken in part from *Cancers* **2022**, *14*, 2178 [146].

## Data Availability

All data needed to evaluate the paper’s conclusions are present. The preclinical datasets generated and analyzed during the current study are not publicly available but from the corresponding author upon reasonable request. The data will be provided following the review and approval of a research proposal, Statistical Analysis Plan, and execution of a Data Sharing Agreement. The data will be accessible for twelve months for approved requests, with possible extensions considered; for more information on the process or to submit a request, contact szewczuk@queensu.ca or wharless@encyt.net.

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
