# Peer review of "Contemporaneous Perioperative Inflammatory and Angiogenic Cytokine Profiles of Surgical Breast, Colorectal, and Prostate Cancer Patients: Clinical Implications"

_cells, 2023, doi:10.3390/cells12232767_

Round 1
Reviewer 1 Report
Comments and Suggestions for Authors
Authors of the submitted manuscript measured 19 cytokines, chemokines and growth factors, primarily involved in inflammation, wound healing and vascularization, in plasma samples obtained from patients (68) after surgery due to breast, colorectal or prostate cancer. Several samples were collected during a 4-week period and dynamics of their elevation or reduction determined. Most of them returned to preoperative values within few days. Authors concluded that due to these changes, which can promote tumor cell growth, such patients have narrow therapeutic window to prevent tumor recurrence.
Authors wrote extensive Introduction and Discussion mentioning known roles of these molecules in tissue repair, wound healing and cancer. However, some of these molecules change after surgery also due to other factors which are not directly related to possible tumor recurrence: pain, massiveness (also acknowledged by authors in the Discussion) or duration of the intervention, reduced nutrition, metabolic changes, infection, possibly type of anesthesia. A number of other molecules change due to surgery causing chain reaction and alteration of other molecules (e.g. cortisol, adrenaline). Certainly there are feeding difficulties after colorectal cancer resection and IGF-1 is expected to decrease after GIT interventions. Likewise, severe changes in nutrition and catabolism are expected to affect liver and its molecules. So, authors should take into consideration such factors as well and write a more balanced text. The existing information on the involvement of the examined 19 molecules should be reduced and more focused.
In Results section, authors have recorded certain effects only in some of the patient groups not all – the number of patients/samples is too small to enable such stratification and tumor-specific conclusions (i.e. just 11 prostate cancer cases). Figures show that values of some markers are widely scattered within a population both before and particularly after the surgery, indicating great inter-individual variability. This scattering may be a cause of the existence of statistically significant difference in certain time points with no difference in the point between them (e.g. Figure 2 A, 14 A, 16 A). More critical discussion is needed. Perhaps it would be insightful to correlate type and massiveness of the surgery (e.g. laparoscopy or open-surgery, partial tissue removal or mastectomy) with concentrations of some of the measured parameters. There is no need to elaborate on mechanisms of action of some of these factors in Discussion – that information can be just referenced.
The article suggests a risk of initiation of tumor recurrence due to post-operative events and points to better approach in perioperative and adjuvant therapies to reduce that risk. Have authors followed-up their patients within a regular control period? How many of them had tumor recurrence? Also, all molecular changes were attributed to the surgery of the tumor type of a disease – what happens with these 19 molecules after other types of (massive) surgery? Some discussion.
It seems that authors performed measurements with ready-made assay components (not counting disolving) and followed the procedure as described in a manuel. If so, there is no need to give so much routine technical details in the Materials and Methods section. It is sufficient to name tests, a producer, and to state that the analysis was done according to the producer's instructions.
Author Response
REVIEWER #1
Comments and Suggestions for Authors
Authors of the submitted manuscript measured 19 cytokines, chemokines and growth factors, primarily involved in inflammation, wound healing and vascularization, in plasma samples obtained from patients (68) after surgery due to breast, colorectal or prostate cancer. Several samples were collected during a 4-week period and dynamics of their elevation or reduction determined. Most of them returned to preoperative values within few days. Authors concluded that due to these changes, which can promote tumor cell growth, such patients have narrow therapeutic window to prevent tumor recurrence.
Authors wrote extensive Introduction and Discussion mentioning known roles of these molecules in tissue repair, wound healing and cancer. However, some of these molecules change after surgery also due to other factors which are not directly related to possible tumor recurrence: pain, massiveness (also acknowledged by authors in the Discussion) or duration of the intervention, reduced nutrition, metabolic changes, infection, possibly type of anesthesia. A number of other molecules change due to surgery causing chain reaction and alteration of other molecules (e.g. cortisol, adrenaline). Certainly there are feeding difficulties after colorectal cancer resection and IGF-1 is expected to decrease after GIT interventions. Likewise, severe changes in nutrition and catabolism are expected to affect liver and its molecules. So, authors should take into consideration such factors as well and write a more balanced text. The existing information on the involvement of the examined 19 molecules should be reduced and more focused.
Author response: Thank you for taking the time to review our paper and for your comments. Regarding the impact of factors not directly associated with tumor recurrence (metabolic changes, anesthesia, hormonal changes during surgery, etc.), we included these considerations in the newly added “limitations” section at the end of the revised manuscript, highlighting that their involvement may impact our findings. Moreover, we included the follow-up information and potential of tumor recurrence in this section as well. The impact of the severity of surgery (laparoscopic versus open) and other factors that can alter inter-individual cytokine level variability was extensively researched and included at the end of the discussion.
With regard to the comment, “The existing information on the involvement of the examined 19 molecules should be reduced and more focused,” it is important to illustrate the differences of cytokine profiling postoperative with the different cancers. To highlight the contemporaneous cytokine profiling in our studies, we have included a graphical abstract to illustrate the landscape of postoperative profiling of cytokines that may contribute to later metastatic disease, which is highly influenced within a short time frame after surgery. This shows how tumor surgery disrupts equilibrium between pro-inflammation and anti-inflammation. This will also help in the treatment after surgery of tumors.
In the Results section, authors have recorded certain effects only in some of the patient groups not all – the number of patients/samples is too small to enable such stratification and tumor-specific conclusions (i.e. just 11 prostate cancer cases). Figures show that values of some markers are widely scattered within a population both before and particularly after the surgery, indicating great inter-individual variability. This scattering may be a cause of the existence of statistically significant difference in certain time points with no difference in the point between them (e.g. Figure 2 A, 14 A, 16 A). More critical discussion is needed. Perhaps it would be insightful to correlate type and massiveness of the surgery (e.g. laparoscopy or open-surgery, partial tissue removal or mastectomy) with concentrations of some of the measured parameters. There is no need to elaborate on mechanisms of action of some of these factors in Discussion – that information can be just referenced.
Author response: Thank you for this comment and suggestions. This is an important point made to explain the inter-individual variability which may be due to the type of surgery and possible mechanism of action. We have included in the revised manuscript a discussion on the impact of the severity of surgery (laparoscopic versus open) and other factors that can alter inter-individual cytokine level variability that was extensively researched by other investigators and included these issues at the end of the discussion.
The article suggests a risk of initiation of tumor recurrence due to post-operative events and points to better approach in perioperative and adjuvant therapies to reduce that risk. Have authors followed-up their patients within a regular control period? How many of them had tumor recurrence? Also, all molecular changes were attributed to the surgery of the tumor type of a disease – what happens with these 19 molecules after other types of (massive) surgery? Some discussion.
Author response: Thank you for this comment and suggestions. Considering the last comment, we have included a discussion on the impact of the severity of surgery (laparoscopic versus open) and other factors that can alter inter-individual cytokine level variability that was extensively researched by other investigators and included these issues at the end of the discussion.
On the comment on follow-up of the patients, Dr. Harless who was in charge of the patients has left the hospital for a position in the USA. He has kept the initial records of the patients. We have heard that some of the patients have passed from developing metastatic disease. It is noteworthy that some patients post-surgery can survive for years but may succumb to metastatic disease later in life. This is an important issue to consider in any studies on post-surgery of cancer patients.
It seems that authors performed measurements with ready-made assay components (not counting disolving) and followed the procedure as described in a manuel. If so, there is no need to give so much routine technical details in the Materials and Methods section. It is sufficient to name tests, a producer, and to state that the analysis was done according to the producer's instructions.
Author response: Thank you for this comment and suggestions. For any cytokine profiling obtained from patient’s plasma, it is important to provide clear details of the proper controls and dilutions made for consistent results. As requested, we have shortened the technical details in the M&M section and named the source.
Submission Date
08 November 2023
Date of this review
18 Nov 2023 21:30:42
Reviewer 2 Report
Comments and Suggestions for Authors
1. It is better not to have abbreviations if not used frequently in the Abstract like HGF). The purpose of the Abstract is to let the reader know simply and clearly the topic of the article.
2. The Conclusion section should be strengthened: the important results and main conclusions drawn in this paper should be highlighted and presented in more precise language.
3. Antitumor background can be strengthened by citing 10.1021/acsbiomaterials.5b00346; 10.1039/C5TB00264H.
4. There are some formatting errors in the article. For example, spelling of references must be checked to meet the journal style (such as Reference 12). Please check carefully and use abbreviation properly.
5. There are 19 Figures in this article. There seem to be too many diagrams in this manuscript. The author could try to change the number of pictures with larger or streamlined pictures.
Comments on the Quality of English LanguageN/A
Author Response
REVIEWER #2
Comments and Suggestions for Authors
- It is better not to have abbreviations if not used frequently in the Abstract like HGF). The purpose of the Abstract is to let the reader know simply and clearly the topic of the article.
Author response: Thank you for this comment and suggestions. Regarding the abbreviations in the abstract, we believe they are necessary as many of these molecules are more commonly known by their abbreviations. Moreover, it is common practice to include these cytokine abbreviations in abstracts, therefore we believe their presence makes it a more comprehensive read.
- The Conclusion section should be strengthened: the important results and main conclusions drawn in this paper should be highlighted and presented in more precise language.
Author response: Thank you for this comment and suggestions. The conclusion has been updated to include the key points of the paper.
- Antitumor background can be strengthened by citing 10.1021/acsbiomaterials.5b00346; 10.1039/C5TB00264H.
Author response: Thank you for this comment and suggestions. The paper you have listed as the antitumor background was very interesting, however, I don’t believe it could be linked to our paper.
- 1021/acsbiomaterials.5b00346 - Fabrication and Characterization of a Novel Anticancer Drug Delivery System: Salecan/Poly(methacrylic acid) Semi-interpenetrating Polymer Network Hydrogel which has no link to our study.
- 1039/C5TB00264H -pH Sensitive Hydrogel Based Acrylic Acid for Controlled Drug Release
- There are some formatting errors in the article. For example, spelling of references must be checked to meet the journal style (such as Reference 12). Please check carefully and use abbreviation properly.
Author response: Thank you for this comment and suggestions. Our referencing is done automatically using the MDPI style of EndNote. The 12th reference mentioned is as it is written exactly in the journal website.
- There are 19 Figures in this article. There seem to be too many diagrams in this manuscript. The author could try to change the number of pictures with larger or streamlined pictures.
Author response: Thank you for this comment and suggestions. The issue regarding the abundance of figures is one we have also debated. However, removing any of the figures or parts of the figures would remove a valuable chunk of the information we present in the paper. It is important to illustrate the differences of cytokine profiling postoperative with the different cancers. To highlight the contemporaneous cytokine profiling in our studies, we have included a graphical abstract to illustrate the landscape of postoperative profiling of cytokines that may contribute to later metastatic disease, which is highly influenced within a short time frame after surgery.
Comments on the Quality of English Language: N/A
Submission Date
08 November 2023
Date of this review
13 Nov 2023 11:39:26
Reviewer 3 Report
Comments and Suggestions for Authors
The article titled “ Contemporaneous Perioperative Inflammatory and Angiogenic Cytokine Profiles of Surgical Breast, Colorectal and Prostate Cancer Patients: Clinical Implications” by Leili Baghaie et.al is well thought, designed and executed. This topic is a general interest for researchers and readers. This also shows that how tumor surgery disrupts equilibrium between pro inflammation and anti inflammation.this also helps in the treatment after surgery of tumors.
Best
Ravi
Author Response
REVIEWER #3
Comments and Suggestions for Authors
The article titled “Contemporaneous Perioperative Inflammatory and Angiogenic Cytokine Profiles of Surgical Breast, Colorectal and Prostate Cancer Patients: Clinical Implications” by Leili Baghaie et.al is well thought, designed and executed. This topic is a general interest for researchers and readers. This also shows that how tumor surgery disrupts equilibrium between pro inflammation and anti inflammation. This also helps in the treatment after surgery of tumors.
Author response: Thank you for this comment and support.
Round 2
Reviewer 1 Report
Comments and Suggestions for Authors
Authors have addressed all my comments and suggestions adequately.